# Novel real number representations in Ising machines and performance evaluation: Combinatorial random number sum and constant division

**Katsuhiro Endo[1,2,3], Yoshiki Matsuda[4,5], Shu Tanaka[2,5,6,7], Mayu Muramatsu[2,8]\***

**1** Research Center for Computational Design of Advanced Functional Materials, National Institute of Advanced Industrial Science and Technology (AIST), Tsukuba, Ibaraki Japan, **2** Quantum Computing Center, Keio University, Yokohama, Kanagawa, Japan, **3** Graduate School of Science and Technology, Keio University, Yokohama, Kanagawa, Japan, **4** Fixstars, Tokyo, Japan, **5** Green Computing System Research Organization, Waseda University, Tokyo, Japan, **6** Department of Applied Physics and Physico-Informatics, Keio University, Yokohama, Kanagawa, Japan, **7** Human Biology-Microbiome-Quantum Research Center (WPI-Bio2Q), Keio University, Tokyo, Japan, **8** Department of Mechanical Engineering, Keio University, Yokohama, Kanagawa, Japan

\* muramatsu@mech.keio.ac.jp

**Data Availability Statement:** All relevant data are publicly available from the Github repository (http://github.com/mmc-research-group/Real-number).

## Abstract

Quantum annealing machines are next-generation computers for solving combinatorial optimization problems. Although physical simulations are one of the most promising applications of quantum annealing machines, a method how to embed the target problem into the machines has not been developed except for certain simple examples. In this study, we focus on a method of representing real numbers using binary variables, or quantum bits. One of the most important problems for conducting physical simulation by quantum annealing machines is how to represent the real number with quantum bits. The variables in physical simulations are often represented by real numbers but real numbers must be represented by a combination of binary variables in quantum annealing, such as quadratic unconstrained binary optimization (QUBO). Conventionally, real numbers have been represented by assigning each digit of their binary number representation to a binary variable. Considering the classical annealing point of view, we noticed that when real numbers are represented in binary numbers, there are numbers that can only be reached by inverting several bits simultaneously under the restriction of not increasing a given Hamiltonian, which makes the optimization very difficult. In this work, we propose three new types of real number representation and compared these representations under the problem of solving linear equations. As a result, we found experimentally that the accuracy of the solution varies significantly depending on how the real numbers are represented. We also found that the most appropriate representation depends on the size and difficulty of the problem to be solved and that these differences show a consistent trend for two annealing solvers. Finally, we explain the reasons for these differences using simple models, the minimum required number of simultaneous bit flips, one-way probabilistic bit-flip energy minimization, and simulation of ideal quantum annealing machine.

**Funding:** This work was supported by the Council for Science, Technology and Innovation (CSTI), Cross-ministerial Strategic Innovation Promotion Program (SIP), Materials Integration for Revolutionary Design System of Structural Materials" (Funding agency: JST) and "Photonics and Quantum Technology for Society 5.0" (Funding agency: QST), and JST COI-NEXT Grant Number JPMJPF2221. This work was supported by JSPS KAKENHI (Grant Numbers JP21K03391, JP23H05447) and JST Grant Number JPMJPF2221. The Human Biology-Microbiome-Quantum Research Center (Bio2Q) is supported by the World Premier International Research Center Initiative (WPI), MEXT, Japan. This work was supported by JST FOREST Program (Grant Number JPMJFR212K, Japan).

**Competing interests:** The authors have declared that no competing interests exist.

# Introduction

Next-generation accelerators including quantum computers are steadily developing. In particular, quantum annealing machines [1–6] can search for the minimum value of the objective function at high speed [7]. Quantum annealing has been studied from the viewpoints of algorithms [8–17] and hardware [18–20]. They are often applied to combinational optimization problems [21] involving the objective functions of the target such as community detection [22], the traveling-salesman problem [23], matrix factorization [24–28] and so on [29–33]. Since some physical phenomena are simulated to find minimum energy states, quantum acceleration of physical simulations can be expected by giving the energy function as the objective function to the quantum solvers [34–37].

A quantum annealing machine specializes in combinatorial optimization problems. The principle of operation is to efficiently find the states in which the value of the objective function is a global minimum using quantum fluctuations. Since the Ising models are used as the basis, they target objective functions with a combination of binary variables as the decision variables.

Physical simulations are one of the most promising practical applications because of their affinity with quantum annealing. This is because many differential equations are derived from the energy functional based on the principle of the least action, and equilibrium states are calculated so that the energy of the system is minimized. Regarding the energy of the system as an objective function, the physical problem of finding the equilibrium state of the system could be solved by a quantum annealing machine.

However, to apply a quantum annealing machine to a physical simulation problem, the state variables need to be transformed into a form that can be represented by the machine. This is because the state variables are real numbers in physical simulations of most engineering problems, which cannot be represented directly by a machine.

Some methods of solving a system of linear equations involving real numbers by quantum circuits have been studied [38–41]. Methods of solving linear problems using quantum annealing as well have also been developed [42, 43]. Pollachini et al. [44] studied how to solve a two-dimensional heat problem involving real numbers by using binary variables. Chang et al. [45] and Dridi and Alghassi [46] studied methods of solving simultaneous linear equations and investigated the performance of their proposed methods.

In order to handle problems involving real numbers by a quantum annealing machine, the conventional method above uses binary number representation, in which one real number is represented as binary number and a quantum bit is assigned to each digit of the binary number. However, we have noticed that using other representations of real numbers can be better than a binary number representation when considering the computational errors in solving problems by the ising machine.

For example, consider solving an optimization problem with only one real number as the decision variable in a classical simulator of annealing machine. Now, the given problem is that the initial value of the real number is 7, and the closer the value is to 8, the better. We also assume that the number is represented as a 4-digit binary integer, i.e., 0111 for 7 and 1000 for 8. The annealing simulator tries to optimize this problem by flipping several bits from state 0111 finally goes to 1000. Considering the path from the initial value to the final state, we find that unless we flip more than two bits at the same time, the state must go through values worse than the initial state. In the case we flip the first and third bits in the first step and then flip the remaining bits, the path is 7 (0111), 13 (1101) and 8 (1000), which contains a worse value (13) than the initial value (7). The more digits, the more bits must be inverted simultaneously to avoid a worse value than the initial value. This means that the annealing simulator needs more

time to overcome unfavorable states for complex real-value problems. A similar property can be inferred for the greedy steepest descent algorithm [47], which is another classical method, because of its repeated local optimization. Although the same trend won't necessarily in quantum annealing, this consideration sufficiently motivates us to consider the representation of real numbers which affects the ease of solving problems.

There are important studies on the problem of such real number representations in the context of evolutionary computing. In this context, the requirement of simultaneous bit flipping of the binary representation mentioned above is known as the Hamming cliff problem [48, 49]. The integer number representation called "Gray coding" has been proposed to eliminate the Hamming cliff problem [50, 51]. Gray coding is a well-constructed representation in which only one bit is flipped to increase a value by one, and this property improves the optimization efficiency of evolutionary computation. However, Gray coding cannot be represented as a weighted linear sum of bits, but it has higher-order nonlinear relations for each bit. In quantum annealing, it is necessary to transform the problem into a QUBO Hamiltonian (quadratic function of bits), and such high-order nonlinear terms cannot be directly embedded. Although the order of the Hamiltonian can be reduced by introducing auxiliary bits, the number of bits required increases and the problem becomes more complicated. Therefore, Gray coding is not suitable for quantum annealing, and we have to discuss real number representations in the range of weighted linear sums. In addition, a previous study of the reverse annealing method [52] has shown that the number of bit flips between the initial state and the optimal state is correlated with the accuracy of the obtained solution, which is also the basis of this study to explore the real number representation. Several methods of representation for multivalued variables have been proposed in previous reports [53–56], showing their advantages and disadvantages for these representations.

Therefore, in this study, we investigate the representation of real numbers using binary variables in annealing machines. We propose three new types of real number representations, i.e., two types of combinatorial random number sums (RANDOM_UNIFORM and RANDOM_-NORMAL) and constant division (CONSTANT) in addition to the conventional binary number representation (BINARY). We investigate their performance by solving the linear $N$-dimensional problem $Ax = b$ with the condition number $k$, which is one of the indicators used to categorize the "difficulty" of linear equations. We find a lot of interesting tendencies. When $N$ and $\kappa$ are small, the representation with binary numbers shows good accuracy, as expected. An increase in the bit number $R$ for the expression of the real number results in a significant increase in accuracy in this case. In problems with a large $N$ or $k$, however, an increase in $R$ in the representation with binary numbers results in decreased accuracy. For problems with a large $N$ or $k$, the representation with constant division unexpectedly shows higher accuracy than the binary representation because of the rapid decrease in the accuracy of the binary representation for a large $N$ or $k$. The representation with constant division shows a monotonically increasing accuracy with $R$. The two representations with random numbers have the advantages of both the binary and constant representations, i.e., they show a similar tendency to the binary representation for small $N$ and $\kappa$ problems and to the constant representation for a large $N$ or $k$. We also determine the reasons for the accuracies of these representations using three toy models. The experiments are performed primarily by a Fixstars Amplify Annealing Engine (denotes Fixstars Amplify AE hereafter) [57]. We also conduct the same experiments using a quantum annealing machine developed by D-Wave Systems [58].

## Methods

### Solving linear equations using Ising machine

We solve the following linear equations of $N$ real variables $\boldsymbol{x} = \{x_0, \ldots, x_{N-1}\}$:

$$\boldsymbol{Ax} = \boldsymbol{b}, \tag{1}$$

where the $N \times N$ real matrix $\boldsymbol{A}$ and $N$-dimensional real vector $\boldsymbol{b}$ are given. To solve the linear equations using an Ising machine, we formulate the linear equations in QUBO models.

QUBO models are generally represented as

$$H = \sum_{0 \leq i,j < M} C_{ij} q_i q_j, \tag{2}$$

where $H$ is a Hamiltonian or an energy function, $\{q_i\}$ are $M$ Ising bits that take only binary values (0 and 1), and $C$ is a real-valued interaction coefficient matrix. Ising machines search the values $\{q_i\}$ so that the Hamiltonian is minimized. The diagonal and off-diagonal elements of $C$ represent the strengths of the bias and quadratic interactions, respectively. We must determine the coefficient matrix $C$ depending on the problem to solve.

In the linear equations, the variables to be found $\boldsymbol{x}$ are real-valued; thus, we need representations of real numbers using only binary variables $\{q_i\}$. Chang et al. [45] used the binary digits mappings

$$x_i = t_i + s_i \sum_{r=0}^{R-1} 2^{-r} q_{Ri+r}, \tag{3}$$

where $R$ is the number of Ising bits used per real number and $s_i$, $t_i$ are real-valued coefficients that scale the mapping. We generalize this mapping to explore better mappings as follows:

$$x_i = t_i + s_i \sum_{r=0}^{R-1} F_{i,r} q_{Ri+r}, \tag{4}$$

where $\{F_{i,r}\}$ are generalized mapping coefficients that represent real number $x$ in a different way. As can be seen from this equation, the number of required Ising bits is $RN$. The specific definition of the mapping coefficients $\{F_{i,r}\}$ is given later.

Once we have the mappings of the real variables $\boldsymbol{x}$, the QUBO models used to solve the linear equations can be easily constructed. Since our goal is to make the linear transformation of $\boldsymbol{x}$ by $\boldsymbol{A}$ as close to $\boldsymbol{b}$ as possible, we take the squared error between $\boldsymbol{A}\boldsymbol{x}$ and $\boldsymbol{b}$ and obtain the Hamiltonian

$$H = \sum_{i=0}^{N-1} \left( b_i - \sum_{j=0}^{N-1} A_{ij} x_j \right)^2 = \sum_{i=0}^{N-1} \left( b_i - \sum_{j=0}^{N-1} A_{ij} \left( t_j + s_j \sum_{r=0}^{R-1} F_{j,r} q_{Rj+r} \right) \right)^2. \tag{5}$$

This Hamiltonian contains only up to quadratic terms; thus, expanding the expression immediately leads to the desired QUBO models. This Hamiltonian takes different forms when different real number representations are used. That is, different real number representations transform the optimization problem in Eq (1) into different types of discrete optimization problems. Note that we obtain exact solutions of the linear equations if and only if the Hamiltonian is zero.

Then, the variable assignment that minimizes the value of the Hamiltonian is computed using Ising machines. The output of the Ising machine is finally transformed into a real value solution of the original problem in Eq (1) on the basis of the real number representation used.

All of the above methods are summarized in Fig 1. In ideal quantum annealers, according to the quantum adiabatic theorem, the optimal state is always obtained if the annealing process takes a sufficiently long time. Thus, we can consider that in an ideal noiseless quantum computer, the longer the annealing time, the better the quality of the solution. Therefore, it is necessary to consider the timeout time $T$ as a parameter that controls the accuracy of the solutions.

## Constructing linear equations to solve

We perform experiments on solving linear equations using an Ising machine. In order to investigate the effect of differences in the difficulty of the linear equations, we use two indicators, the number of variables, $N$, and the condition number $\kappa$ of matrix $\boldsymbol{A}$, to measure such difficulty. As the number of variables increases, the difficulty is expected to increase because the number of equations to be satisfied also increases. $\kappa$ is defined as the ratio of the maximum and minimum singular values,

$$\kappa(\boldsymbol{A}) = \frac{\sigma_{\max}(\boldsymbol{A})}{\sigma_{\min}(\boldsymbol{A})}. \tag{6}$$

This condition number $\kappa$ measures the ratio of changes in the solutions $\boldsymbol{x}$ relative to the change in the term $\boldsymbol{b}$, and thus large $\kappa$ also corresponds to high difficulty.

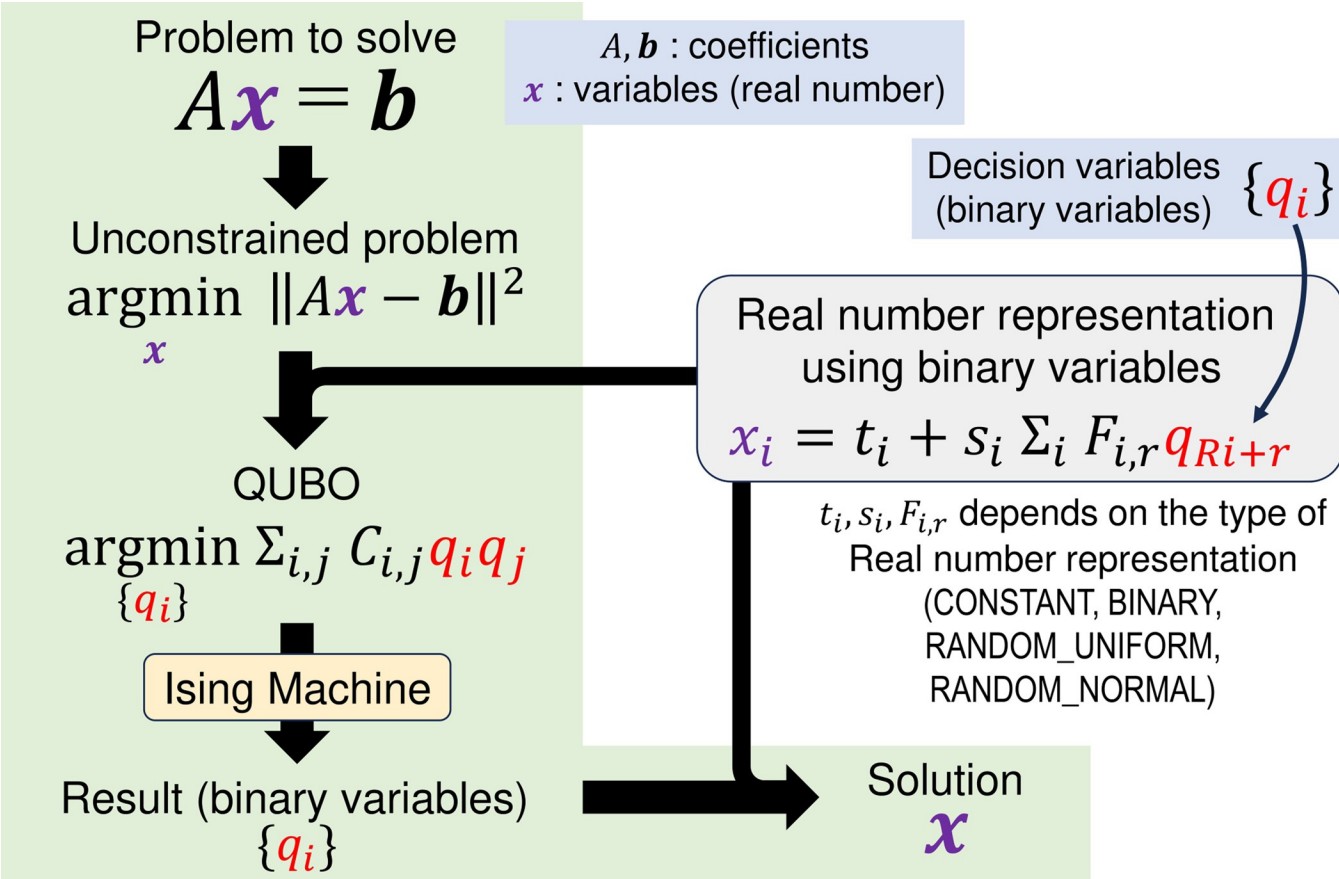

**Fig 1. Schematic diagram showing the process of solving the linear equations by Ising machines.**

To generate matrix $A$ with a certain condition number $k$, we use the following procedure: First, we generate random matrix $M$ whose $N \times N$ elements are sampled from independent standard normal distributions. This matrix $M$ does not have the given condition number $k$. Second, we decompose the matrix $M$ to $M = QR$ by QR-decomposition. To control singular values, we ignore the matrix $R$ and use a diagonal matrix $\Lambda$. Third, the diagonal elements of $\Lambda$ are sampled from the uniform distribution range -1 to 1 and linearly rescaled to make the maximum and minimum absolute values of the elements equal to $\kappa$ and 1, respectively. Finally, we generate matrix $A = Q^T \Lambda Q$ with the condition number $k$.

Next, we generate the vector $b$, by generating the solution vector $x_*$ sampled from the independent uniform distribution range -1 to 1, then calculating the vector $b$ from the linear equations $b = Ax_*$. This guarantees that all the elements of the solution are in the range -1 to 1.

## List of real number representations to be compared

In this work, the following four real number representations are used for comparison by setting $F_{i,r}, s_i, t_i$ in Eq (4):

1. RANDOM_UNIFORM

$$F_{i,r} \sim \mathcal{U}(0,1), s_i = \frac{2}{\Sigma_r F_{i,r}}, t_i = -1, \tag{7}$$

where $U(0,1)$ is the uniform distribution in the range 0 to 1.

2. RANDOM_NORMAL

$$F_{i,r} \sim \mathcal{N}^+\left(0, 1^2\right), s_i = \frac{2}{\Sigma_r F_{i,r}}, t_i = -1, \tag{8}$$

where $N^+(\mu, \sigma^2)$ is a truncated normal distribution with mean $\mu$ and variance $\sigma^2$, and is limited to positive value.

3. CONSTANT

$$F_{i,r} = \frac{1}{R}, s_i = 2, t_i = -1. \tag{9}$$

4. BINARY

$$F_{i,r} = 2^{-r}, s_i = 2, t_i = -1. \tag{10}$$

RANDOM_UNIFORM and RANDOM_NORMAL have randomness in representing real numbers. Note that there is a previous study of real number representations by summing random numbers in the field of evolutionary algorithms [59]. CONSTANT and BINARY have no randomness. In all these representations, $s_i$ and $t_i$ are series of values that make the variable $x$ represent real numbers in range -1 to 1.

## Error criteria

To validate the solutions obtained using Ising machines, we use the dimension-normalized absolute error $E$ given by

$$E = \frac{|\boldsymbol{x}_* - \boldsymbol{x}|}{\sqrt{N}}, \tag{11}$$

where $\boldsymbol{x}_*$ is the optimal solution and $\boldsymbol{x}$ is the obtained result. The normalizing factor in the denominator is set so that the maximum error is 2. The case of the maximum error occurs when the optimal and obtained results become $\boldsymbol{1}$ and $-\boldsymbol{1}$, respectively, or vice versa, because all the elements in both vectors range from -1 to 1. Here, $\boldsymbol{1}$ denotes the vector in which all elements are 1.

We do not use the relative error but the absolute error for the following reason. Since the smallest difference among the possible values in these representations is fixed, the absolute error does not depend on the norm of the optimal solution; thus, we would overestimate the error using the relative error. For example, when the possible values are {-1.0, -0.9, -0.8, . . ., 0.9, 1.0} and we consider the optimal solutions 0.04 and 0.94, the smallest absolute error is 4% for both solutions, but the relative error for these solutions becomes 100% and 4.4%, respectively. Thus, the relative error is not suitable for error evaluation.

## Error factor

To systematically analyze our simulation results, we group the average errors of conditions that differ only by the number of Ising bits per real number, $R$, as the error vectors $\boldsymbol{y}$, and embed these vectors in a two-dimensional space to visualize them. Each vector $\boldsymbol{y}$ has four elements, and the first, second, third, and fourth elements correspond to $R = 2, 4, 8$ and $16$, respectively. We have different error vectors corresponding to the different combinations of the method, number of variables, $N$, condition number $k$, and timeout time $T$. For visualization, we factorize the vector $\boldsymbol{y}$ into

$$\log \boldsymbol{y}^i = \bar{y}\boldsymbol{1} + a_1^i \boldsymbol{p}_1 + a_2^i \boldsymbol{p}_2 + \epsilon^i, \tag{12}$$

where $\boldsymbol{y}^i$ is the error vector of the $i$th condition, $\bar{y}$ is the average logarithmic error of all conditions, $\boldsymbol{p}_1$ and $\boldsymbol{p}_2$ are the first and second-factor vectors, respectively, $a_1^i$ and $a_2^i$ are the coefficients of each factor vector for the $i$th error vector, and $\epsilon^i$ is the $i$th factorization error vector.

To determine the factor vectors and all the coefficients, we use singular value decomposition. To obtain a low-rank approximation of the data matrix $\{\boldsymbol{y}^i\}$, $\boldsymbol{y}^i$ is decomposed as

$$log \, \boldsymbol{y}^i - \bar{y}\boldsymbol{1} = \sum_{k=1}^{4} u_k^i \sigma_k \boldsymbol{v}_k, \tag{13}$$

where $\sigma_k$ is the $k$th largest singular value and $u_k^i$ and $\boldsymbol{v}_k$ correspond to the left and right singular vectors, respectively. From this result, we select the factor vectors and their coefficients as

$$a_k^i = u_k^i, \boldsymbol{p}_k = \sigma_k \boldsymbol{v}_k (\text{for } k = 1, 2). \tag{14}$$

## Results

### Calculation results and qualitative trends

To evaluate the performance of the four representations, the RANDOM_UNIFORM, RANDOM_NORMAL, CONSTANT and BINARY, we perform the experiments denoted in the section on solving linear equations using an Ising machine with various settings. In the experiment, we used the Fixstars Amplify Annealing Engine, a GPU-based simulated annealing machine, and the Amplify SDK middleware developed by Fixstars Amplify. The parameters changed are the number of variables, $N$, (4, 8, 16, 32, 64, 128, 256), the condition number $\kappa$ of linear transformation matrix $A$ (1, 16, 256), the number of Ising bits per real number, $R$, (2, 4, 8, 16), and the timeout time $T$ (100 ms, 1000 ms, 10000 ms). All of these combinations are tested for all representations, and 16 independent experiments are performed for each combination, giving 16,128 (= 16×4×7×3×4×3) results for solving linear equations. Since the generation of the matrix $A$ and the vector $b$ and the Ising machine itself have randomness, the results for the same parameters are completely different in each run.

We plot the selected results of these runs in Fig 2. The vertical axis represents the average error $E$ of 16 trials and the horizontal axis and subplots represent different conditions. From this figure, we can see several qualitative trends. First, for small $N$ and small $k$, BINARY gives the most accurate solution. This result is in line with the usual intuition that binary notation can efficiently represent real numbers. However, with increasing $N$ or $k$, the performance of BINARY deteriorates, despite the increased number of Ising bits, $R$. This trend cannot be seen in the other three representations; thus, BINARY should not be used for large or difficult problems. Next, the performance of CONSTANT increases monotonically with an increase of $R$ in all situations. Although its performance is inferior to that of the other three methods on average, it is relatively high for large $N$ and large $k$.

Finally, RANDOM_UNIFORM and RANDOM_NORMAL (denoted RANDOMs hereafter) have the advantages of both BINARY and CONSTANT. RANDOMs show similar performance to BINARY for small $N$ and small $k$, and similar performance to CONSTANT for large $N$ and large $k$. For the other conditions (small $N$ and large $k$, or large $N$ and small $k$), RANDOMs show better performance than BINARY and CONSTANT. Note the performance characteristics of the two RANDOM methods themselves are similar for all conditions.

### Error factor visualization of the obtained results

Having found the qualitative performance trends, we next analyze these trends quantitatively. In our experiments, the contribution rate up to the second largest singular value (the ratio of squared singular values) is more than 98.6%. Hence, the data matrix is satisfactorily approximated by using up to the second singular value. We plot the two obtained factor vectors $p_1, p_2$ in different colors in Fig 3A. The value of the first component of the vectors $(p_1)_1, (p_2)_2$ is represented by the first vertical value from the left, and so on. According to this figure, the two vectors seem to encode different features. Each element of the first vector $p_1$ takes a higher value as $R$ increases (blue line). On the other hand, each element of the second vector $p_2$ takes a lower value as $R$ increases (orange line). In other words, these vectors encode two opposite trends: one where the performance worsens as $R$ increases and the other where the performance improves as $R$ increases.

We show scatter plots of the coefficient values $\left(a_1^i, a_2^i\right)$ for all conditions in Fig 3B. In this figure, the methods are distinguished by color. The $x$- and $y$-axis of the plot represent the coefficient values $a_1^i$ and $a_2^i$, respectively; that is, with increasing $R$, an increase in the $x$-axis value indicates an increased error, while an increase in the $y$-axis value corresponds to a decreased

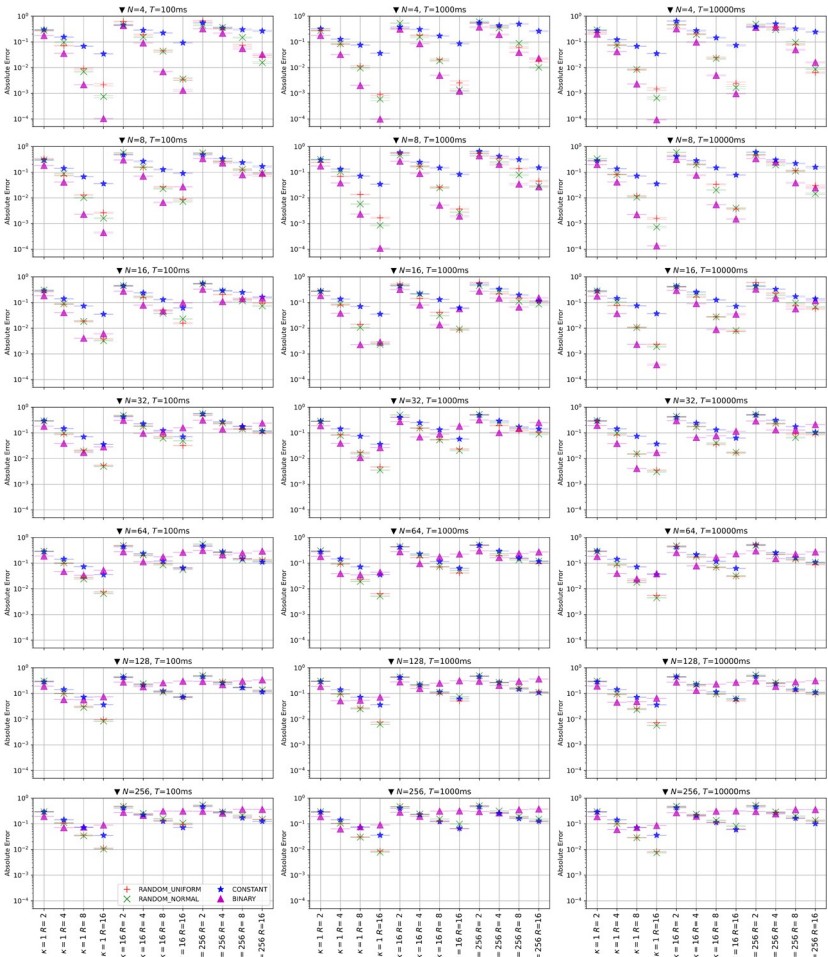

**Fig 2. Mean errors of the four representations, RANDOM_UNIFORM, RANDOM_NORMAL, CONSTANT, and BINARY, when linear equations are solved using the Fixstars Amplify AE with various settings.** The error bars denote the standard error of the estimated mean errors.

error. In addition, the positive value of all elements of $\boldsymbol{p}_1 + \boldsymbol{p}_2$ indicates that the overall performance worsens toward the upper right, regardless of $R$. We also show the color-coded results by condition for each representation in Fig 3C.

From these characteristics, we can quantitatively evaluate the trend of each representation. For CONSTANT, the plot starts from the lower middle, which corresponds to its lower average performance for small and simple problems and extends to the upper right, which corresponds to the monotonic improvement with increasing $R$ for large and complex problems. For BINARY, it starts from the lower-left corner, which corresponds to its superior performance for small and simple problems but extends to the far right more rapidly than for the other representations, which corresponds to the severe performance degradation with increasing $R$ for large and complex problems. For RANDOMs, the plots start near the starting point of the BINARY plot and approach the CONSTANT plot. Thus, we can see from this figure that RANDOMs combine the advantages of BINARY and CONSTANT.

In addition, we conduct the same simulations on a quantum annealing machine developed by D-Wave Systems. Since D-Wave Systems has two types of annealing solver, a hybrid solver and a full quantum solver, we perform the annealing simulations on both solvers. As a hybrid

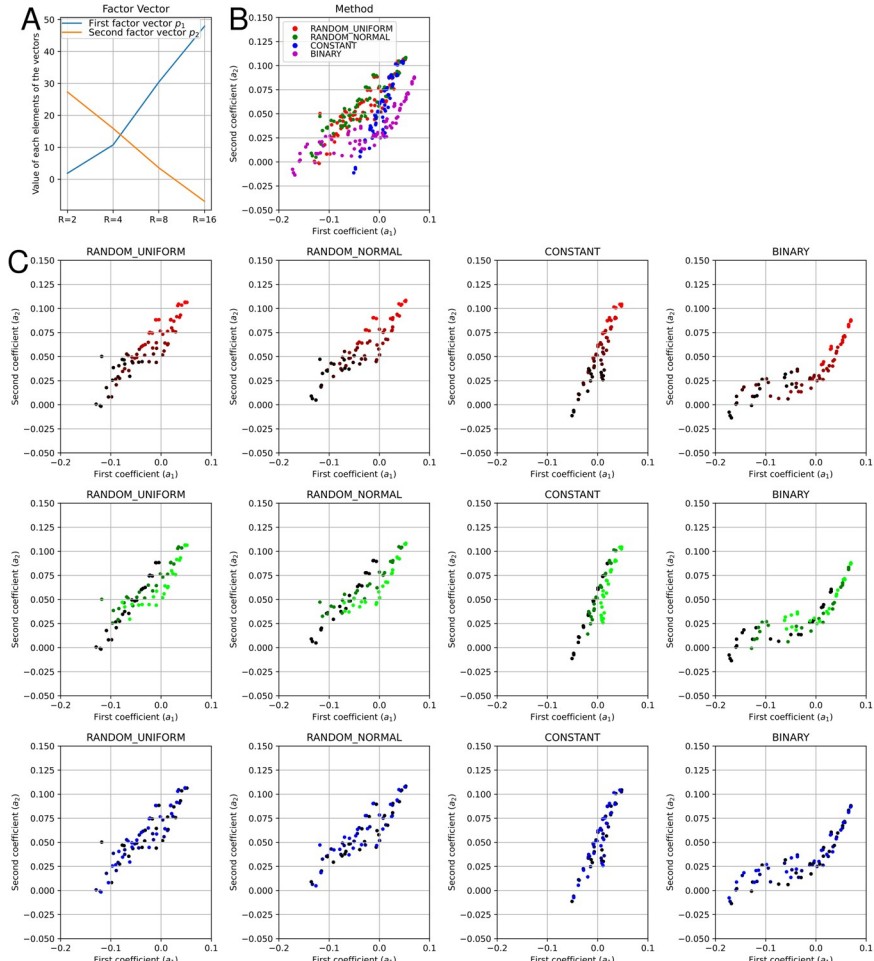

**Fig 3. Plots of the coefficient values $\left(a_1^i, a_2^i\right)$ for various conditions using the Fixstars Amplify AE.** (A) The basis vectors of each coefficient, where the first and second basis vectors $p_1, p_2$ represent the performance worsens as the number of bits $R$ increases and the performance improves as $R$ increases, respectively. (B) The coefficient values of all conditions tested. (C) The coefficient values by real number representation and by parameter. Lighter colors indicate higher parameter values.

solver, we used hybrid_binary_quadratic_model_version2 via the Amplify SDK middleware. As a full quantum solver, we used Advantage_system1.1 via the Amplify SDK middleware. In this case, The Amplify SDK middleware uses find_clique_embedding function provided by D-wave API for minor embedding and the majority vote for the chain break resolution algorithm. In Fig 4, we show the factor visualization results of both solvers, where Fig 4A and 4B show the results for the hybrid and the full quantum solvers, respectively. In the results of the hybrid solver (Fig 4A), we use all combinations of the following conditions: $R$ = 2, 4, 8, 16; $N$ = 4, 16, 64; $\kappa$ = 1, 16, 256; and all four representations with timeout time $T$ = 3000 ms. In the results of full quantum solver (Fig 4B), we use all combinations of the following conditions: $R$ = 2, 4, 8, 16; $N$ = 2, 4, 8, 11; $\kappa$ = 1, 16, 256; and all four representations with the read counts $T_C = 10^3, 10^4$ (corresponding to timeout time $T$). All the plots under different conditions are indicated by the same color depending on four representations of the real number, i.e., RANDOMs, CONSTANT and BINARY. The scale of the problem is limited owing to the hardware limitation (especially in full-quantum solver). Each combination of conditions is employed 8

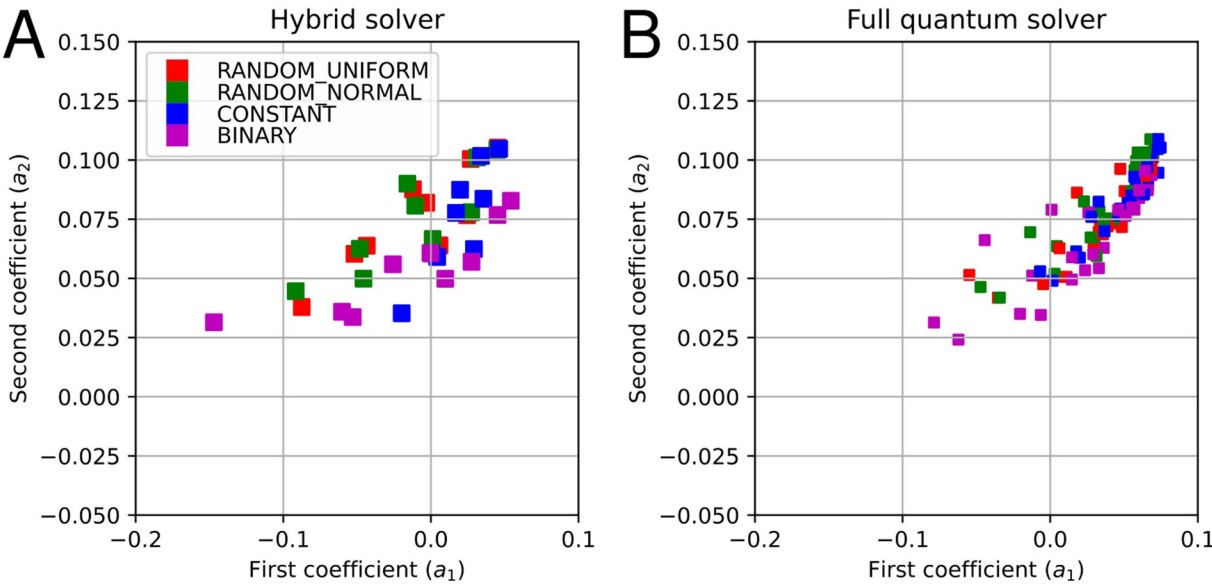

**Fig 4.** Plots of the coefficient values $\left(a_1^i, a_2^i\right)$ for various conditions using (A) D-Wave hybrid solver with timeout time $T = 3000$ ms. and (B) D-Wave full quantum solver with read counts $T_C = 10^3$, $10^4$ (corresponding to timeout time $T$).

times. Note that in the factor visualization plots, we use the same two-factor vectors $(\boldsymbol{p}_1, \boldsymbol{p}_2)$ as for the Fixstars Amplify AE results. From Fig 4A, the results for the hybrid solver show good agreement with the quantitatively evaluated results obtained with the Ising machine; the trend of each representation does not change, and therefore the characteristics are not unique to the Ising machine. For the full quantum solver in Fig 4B, however, there is no significant difference between the methods. This could be due to the insufficient size of the program or the limited precision of the coefficient weights of the D-Wave hardware, including the effect of minor embeddings.

## Discussion of the causes of performance trends

In the above sections, the error tendency of each representation was clarified. In this final section, we discuss the causes of the trends by considering (1) the minimum required number of simultaneous bit flips, (2) one-way probabilistic bit-flip energy minimization and (3) the simulation of ideal quantum annealing machine.

**Task 1** We first note that the energy may not be reduced unless several bits are inverted simultaneously. For example, if we represent an integer as a binary number, and the current value is 7 [0111], we need to flip all four bits simultaneously to reach 8 [1000]. Inverting fewer than 4 bits will result in the same or increased distance from 8. On the other hand, if the number is represented by the sum of each bit, 1-bit inversion is sufficient to reduce the distance to the target in all cases (e.g., 7 [111110111] to 8 [111111111]). In Tasks 1 and 2 we are considering the repeated process of moving to a such neighborhood. The local neighborhood search with Hamming distance of 1 is not a strictly necessary constraint in simulated annealing. However, for example, in the case of the simulated annealing method that utilizes the Metropolis-Hastings algorithm (which is commonly used as an implementation of simulated annealing), if the state is changed drastically at a single step, the energy difference between the states before and after the change becomes too large, resulting in a very low probability of movement. For this reason, it is certain that local neighborhood search, in which only one part of the state is

changed, is often employed for efficiency reasons. We hypothesize that this effect occurred in the real number representations; thus, we conduct an experiment to investigate this hypothesis.

We experimentally investigate how many simultaneous bit flips are required to reach the target value from the initial bit value for a one-dimensional real value. First, the initial bits are set uniformly at random, and the target value is sampled from the uniform distribution $U$ (-1,1). Then, we search for the state that is closest to the target value, limiting the path to only repeated transitions to states with decreasing distances that can be reached by the flipping of bits up to the specified maximum number of simultaneous bit flips. The distance here is the Euclidean distance between the real value which is expressed in the binary variables using the real number representation and the target real value. We performed such simulations using combinations of different numbers of bits $R$ = 2, 4, 8, 16, a maximum number of simultaneous bits flip, $X$, up to $R$, and all four representations. Since the task is stochastic, we run 1,000 independent trials for each condition. Fig 5 shows the results of the task, where each subplot in the figure corresponds to a different condition. The condition of each subplot (the values of $\kappa$ and $R$) is listed in the subtitle at the top of each subplot. The trials in the plots are sorted by the

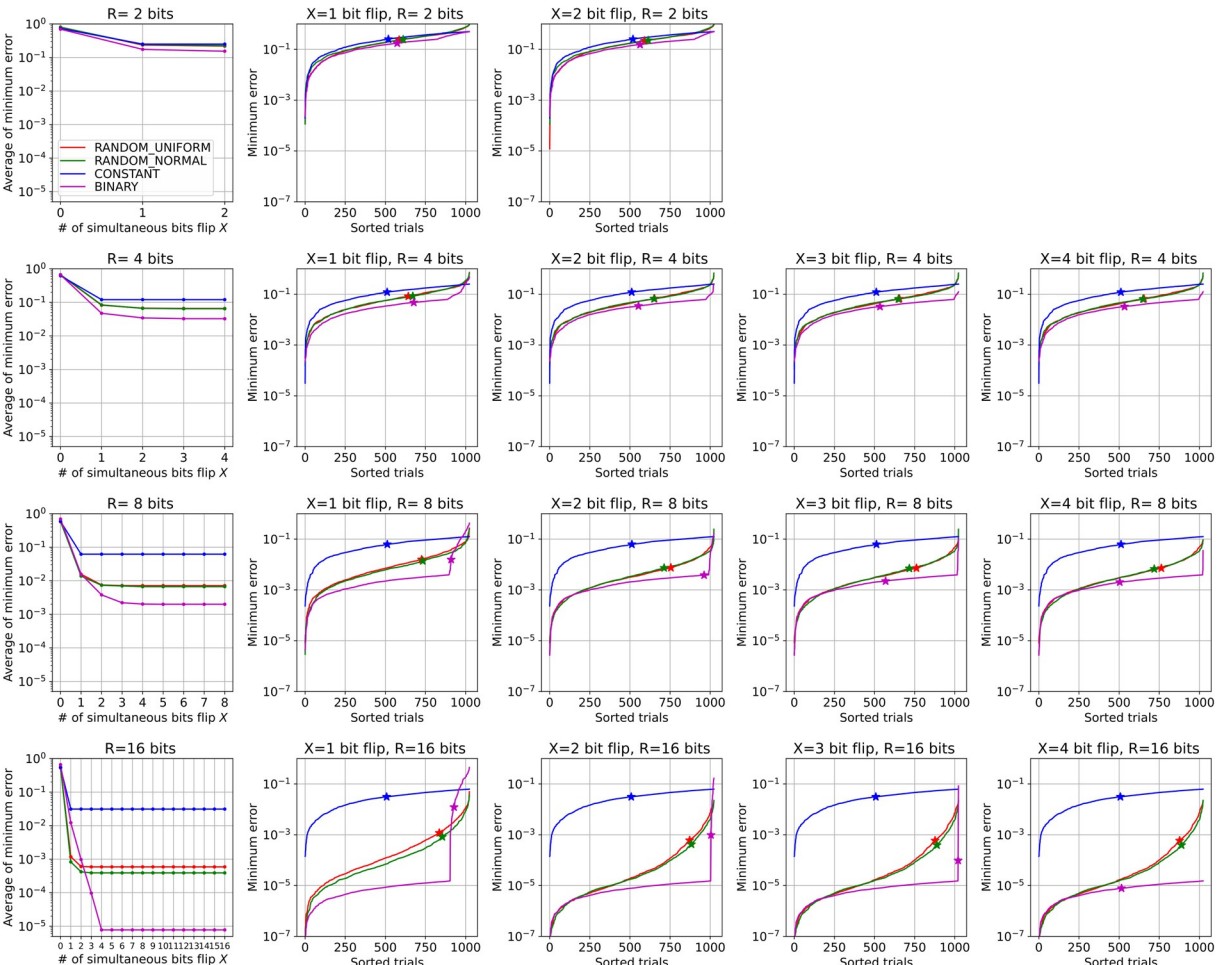

**Fig 5. Results of Task 1.** The plot is divided into subplots vertically by the number of bits, $R$, and horizontally by the maximum number of simultaneous bits flip, $X$. The errors achieved in all trials are sorted in each subplot. The left-most subplot summarizes the averages of these trials.

minimum error that can be reached. The average error of all trials for each condition is summarized in the leftmost column of the figure.

From the figure, we can make some observations. For CONSTANT, one-bit flip is enough to reach the minimum possible error, but its minimum value is the largest among the representations, as shown in the left column. The minimum error is much larger than that for the other representations for large $R$. On the other hand, for BINARY, it seems that $\log_2 R$ simultaneous bit flips are required to reach the minimum possible error. Strange "error bumps" are observed in each trial, especially for large $R$. Although the errors on the low side of the bump are the smallest among the representations, the error grows rapidly on the high side of the bump, as if a phase transition has occurred. This bump causes BINARY to require additional simultaneous bit flips to reduce the error. For RANDOMs, the average error is roughly between those of BINARY and CONSTANT. However, looking closely at the case of $R = 16$, we see that it has the highest performance for a few bit flips ($X = 1, 2$). This is mainly because the error curve is smoothly connected without any strange bumps. More precisely, the lower side of the curve in RANDOMs behaves similarly to the lower side of the bump in BINARY, and the curve smoothly rises to the right, which is why RANDOMs have a superior average performance to BINARY.

**Task 2** In Task 1, we clarified the importance of multiple simultaneous bit flips. In Task 2, further validation is conducted to enforce the importance of our findings from Task 1, by investigating the behavior of bit flips when they are performed stochastically. In Task 1, we assumed that the path to the minimum error is always selected. More realistically, however, such a path is not always achievable. Therefore, in Task 2, we investigate the minimum error that can be reached when bit flips are performed stochastically.

First, similarly to that in Task 1, the initial bits are set uniformly at random, and the target value is sampled from the uniform distribution $U(-1,1)$. Then, we flip each bit with probability $1/R$ independently so that the average number of flips is 1. The stochastic flips are executed simultaneously for all bits in a step, and each flip is accepted as the next state if and only if the error decreases. This step is repeated for the prescribed number of steps. We call this process one-way probabilistic bit-flip energy minimization because the state transitions only in the direction that decreases the error.

We perform these simulations by combining the conditions; the number of bits $R = 2, 4, 8, 16$; the number of steps $\kappa = 10^2, 10^3, 10^4, 10^5$; and all four representations. Since the task is stochastic, we run 10,000 independent trials for each condition. Fig 6 shows the minimum error reached for the task, whose each subplot in the figure corresponds to a different condition. The trials in the plots are sorted by the minimum error reached. The average reached error for all trials of each condition is summarized in the leftmost column of the figure.

From the figure, we can make observations similar to those for Task 1. For CONSTANT, only about $10^2$ steps are required to reach the minimum possible error for all $R$ values, which is unexpectedly small. In contrast, the minimum possible error is the greatest among the representations. On the other hand, BINARY requires a large number of steps to reach the minimum possible error. As shown in the leftmost column in the figure, even $10^5$ steps are not enough to minimize the error for $R = 16$. In the plot for each trial, a sudden change in the slope of the curve is observed. This sudden change is considered to correspond to the "error bumps" in Task 1. Increasing the number of steps (moving to the right in the subplots) causes the bumps in the curves to move to the right, which indicates a transition from the states trapped in the bumps to better states, and a large number of steps are required for all trials to overcome the transition. This property may be the reason why the performance of BINARY does not improve at all even if the number of bits increases. In contrast to BINARY, the error curve of RANDOMs is smooth and no bump is observed in this case. The entire curves shift

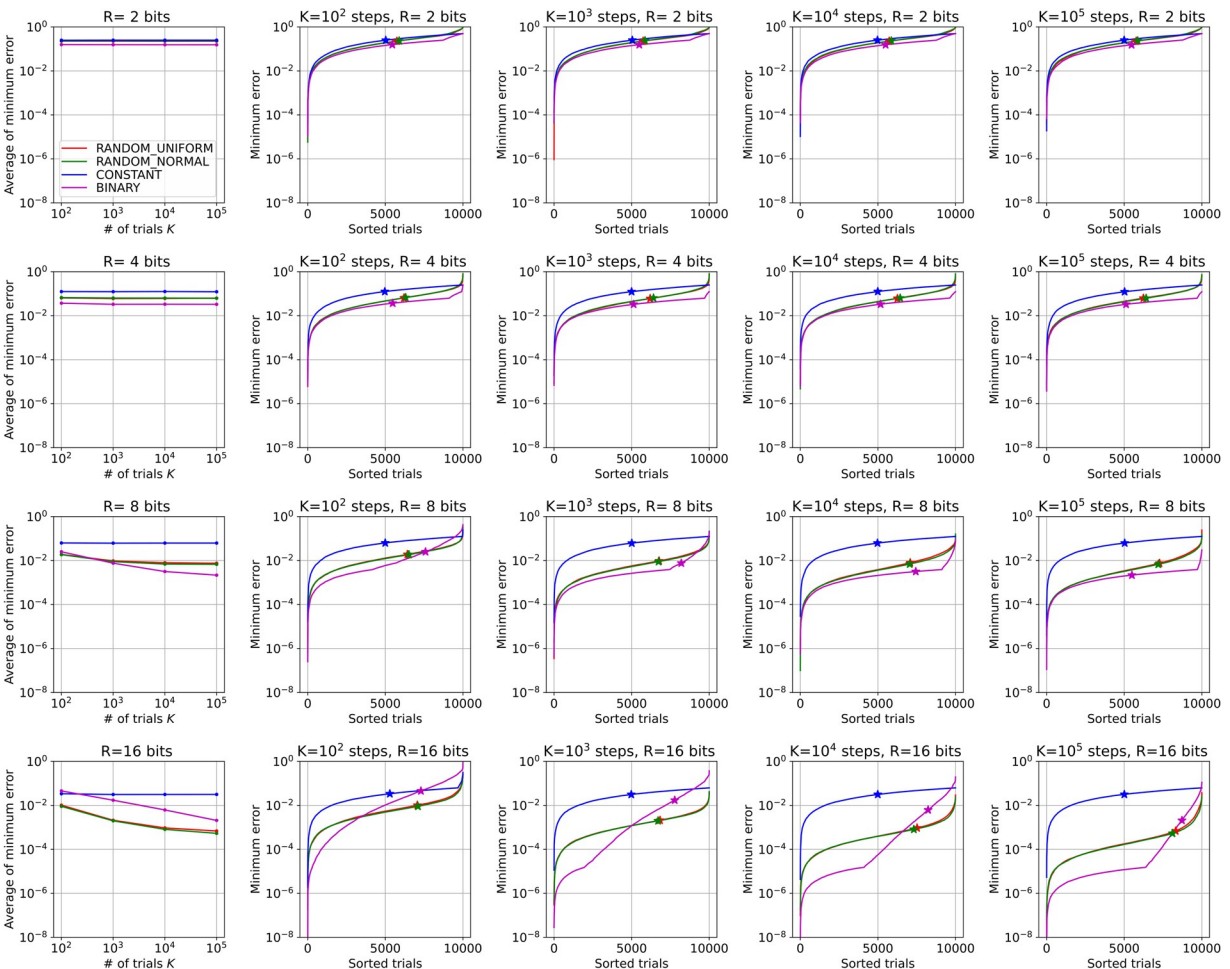

**Fig 6. Results of Task 2.** The plot is divided to subplots vertically by the number of bits, $R$, and horizontally by the number of flip steps, $X$. The errors achieved in all trials are sorted in each subplot. The left-most subplot summarizes the averages of these trials.

uniformly downward, which means that no transition occurs. This difference partly explains why RANDOMs obtain the smallest minimum errors among the representations for $R = 16$.

The two above analyses provide us with an understanding of the factors responsible for the trends in the experimental results for the linear equations. For multidimensional problems, the negative effects of the bumps in BINARY are expected to increase further. The trapping at bumps is considered to occur stochastically; thus, the probability of at least one occurrence of trapping increases when there are multiple real numbers. As a result, increasing the number of dimensions, $N$, of the linear equations will lead to extremely poor results for BINARY.

**Task 3** In Tasks 1 and 2, we examined the efficiency of bit-flip-based search, which mimics the process of simulated annealing with local neighborhood searches when using different real number representations. However, it is not certain that quantum annealing machines have the same properties. Therefore, in Task 3, we discuss the properties of quantum annealing machines by performing a task that directly simulates the process of quantum annealing using a simple optimization problem. Note that such analysis by simulating the annealing process has been done in previous studies, albeit in the context of reverse annealing [60].

In Fig 7, we show the results of solving the following problem by the simulation of quantum annealing. First, we considered a single real number as a decision variable $x$ and a cost function

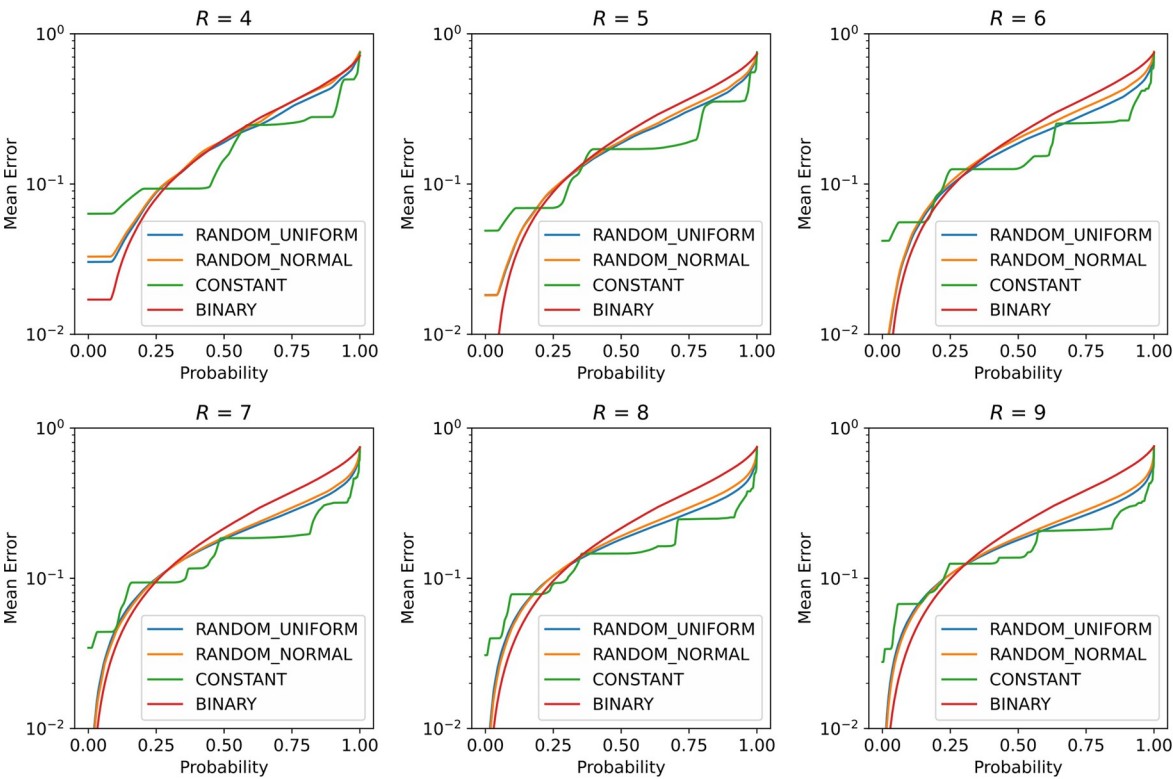

**Fig 7. Results of Task 3.** The figure shows panels for different number of bits, $R$. Each curve indicates the averaged error probability function $\langle E(p) \rangle$.

as the squared distance from a target real value $y$, $L = (x-y)^2$. We assume that the decision variable $x$ ranges from 0 to 1 and that the target real value $y$ is chosen from a uniform distribution in the range from 0 to 1. The optimal value of $x$ for this problem is obviously $x = y$. To solve this problem using quantum annealers, $x$ is represented as $R$ binary variables $\{q_i\}$ ($i = 1,...,R$) using some kind of real number representation ($x = t + s \sum F_i q_i$), and then the cost function takes the QUBO form, $L = (x - y)^2 = (t + s \sum F_i q_i - y)^2$.

To find the optimal value of $x$ of this QUBO problem the following simulation of quantum annealing on classical computers was performed. First, to convert the obtained QUBO Hamiltonian into a quantum Hamiltonian on quantum computers, we replace $q_i$ with $(1-Z_i)/2$ for each binary variable $q_i$ to obtain the quantum Hamiltonian $H_Z$, which is a $2^R \times 2^R$ Hermitian matrix. Here, $Z_i$ is the Pauli $Z$ matrix on the $i$-th qubit. The quantum annealing machine can generate the ground state of this quantum Hamiltonian $H_Z$, which corresponds to the optimal solution of the QUBO problem.

Next, we simulate the quantum annealing process of the quantum annealing machine to find the ground state of this quantum Hamiltonian $H_Z$. We define a time-dependent Hamiltonian $H(t) = (1 - t/T)H_X + (t/T)H_Z$ that varies in time from the transverse field Hamiltonian $H_X = \sum X_i$ to the target Hamiltonian $H_Z$. Here, $X_i$ is the Pauli X matrix on the $i$-th qubit and $T$ is the time taken for the Hamiltonian to change over time. According to the quantum adiabatic theorem, which is the theoretical basis of quantum annealing, a quantum state that was the ground state of the initial Hamiltonian $H(t = 0)$ at the initial time remains in the ground state of the Hamiltonian at each time, $H(t)$, if the Hamiltonian changes with time slowly enough. We therefore simulate the time development of the quantum state $\varphi(t)$ under the time-

dependent Hamiltonian $H(t)$, $d\varphi(t)/dt = -iH(t)\varphi(t)$, with the ground state of the Hamiltonian $H_X$ as the initial quantum state $\varphi(t = 0)$. This simulates the actual computational process in a quantum annealing machine without any noise.

To simulate this quantum time evolution on classical computers, we compute $\varphi(t + \Delta t) = \exp\{-i\Delta t H(t)\}\varphi(t)$ from $t = 0$ to $T$ to obtain the final quantum state $\varphi(T)$, which is one of the explicit finite-difference methods. Here, $\Delta t$ is the time step, and $T = 1.0$ and $T/\Delta t = 10^4$ were used in this experiment, in which we have confirmed that this number of divisions $T/\Delta t$ is sufficient for the calculation accuracy.

Note that when $T$ is sufficiently large, the final state $\varphi(T)$ should coincide with the ground state of the Hamiltonian $H_Z$ from the quantum adiabatic theorem. However, we are not interested in the case where $T$ is sufficiently large. This is because the BINARY representation, which has the highest representational capability for real numbers, should yield the best results if the optimal state of the QUBO problem is obtained. Instead, what we are interested in is the difference in the results that different real number representations yield a realistically small $T$.

If $T$ is not sufficiently large, the accuracy that can realistically be achieved by using each real number representation can be examined. Here, if $T$ is not sufficiently large, the final state $\varphi(T)$ does not coincide with the optimal state but is a superposition of multiple states. When such a final state $\varphi(T)$ is measured, the measurement result and the solution $x$ derived from it vary stochastically. Thus, we can calculate the probability $p$ that the error between the given result $x$ obtained from the bits measured using real number representation and the correct answer $y$, $|x-y|$, is smaller than the given threshold $E$. In other words, the error probability function $E(p)$ can be defined to indicate that the error $E$ can be achieved with the probability $p$. Here, note that the error probability function $E(p)$ also depends on the randomness of the selected the target value $y$ and the randomness of the real number representation. To account for this randomness, the average $E(p)$ for 1000 independent trials, $\langle E(p)\rangle$, is calculated for statistical comparison. This averaged error probability functions $\langle E(p)\rangle$ is plotted in Fig 7, in which the results are shown in different panels for different numbers of binary variables used in the real number representation, $R$. The differences in the real number representation are indicated by different curves in each inset.

Fig 7 shows that the statistical trends of the results differ significantly depending on the real number representation. First, in the BINARY representation, the error is smaller for a small $p$ than in the other real number representations. This result is consistent with the fact that the BINARY representation has the highest representational capability.

On the other hand, for a large $p$ in the BINARY representation, the error deteriorates more rapidly than for the other real number representations, and the error is clearly larger than for the other methods especially when the number of bits used is 6 or more. This result is also consistent with the experimental results (Figs 2–4), which show that the BINARY representation has high representational power but fails to take advantage of that power and loses to other real-number representations when a large number of bits are used.

Next, the CONSTANT representation has the largest error at a low probability $p$ owing to its limited expressive power, but the smallest error is achieved at a high probability $p$. Moreover, the two RANDOM representations are a good mix of the BINARY and CONSTANT representations because they have smaller errors than the BINARY representation at low probabilities and smaller errors than CONSTANT at high probabilities. These results are also consistent with the experimental results.

Therefore, this task shows that as in the case of simulated annealing, the difference in real number representation has also a significant impact on quantum annealing. We believe that these results indicate that the arguments discussed in Tasks 1 and 2 regarding the number of

simultaneous bit flips required to reduce energy in classical simulated annealing are also important in quantum annealing.

## Conclusions

In this study, we proposed three new types of real number representation using binary variables in Ising machines and compared these representations under the problem of solving linear equations. The proposed real number representations were two types of combinatorial random number sum and one type of constant division. With addition of the binary number representation, their performances were investigated by solving the linear equations $Ax = b$ with various conditions. As a result, we found many interesting tendencies for the proposed real number representations in the Ising machines depending on the characteristics of the linear equation, which will be very helpful for the analysis of various engineering problems by the Ising machine.

## Author Contributions

**Conceptualization:** Katsuhiro Endo, Mayu Muramatsu.

**Data curation:** Katsuhiro Endo.

**Formal analysis:** Katsuhiro Endo.

**Funding acquisition:** Mayu Muramatsu.

**Investigation:** Katsuhiro Endo.

**Methodology:** Katsuhiro Endo.

**Project administration:** Katsuhiro Endo.

**Software:** Katsuhiro Endo.

**Supervision:** Yoshiki Matsuda, Shu Tanaka.

**Validation:** Katsuhiro Endo.

**Visualization:** Katsuhiro Endo.

**Writing – original draft:** Katsuhiro Endo, Mayu Muramatsu.

**Writing – review & editing:** Katsuhiro Endo, Yoshiki Matsuda, Shu Tanaka, Mayu Muramatsu.

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
