## [Decision Letter · Decision Letter 0]

12 Sep 2023

PONE-D-23-25182Novel real number representations in Ising machines and performance evaluation: Combinatorial random number sum and constant divisionPLOS ONE

Dear Dr. Muramatsu,

Thank you for submitting your manuscript to PLOS ONE. After careful consideration, we feel that it has merit but does not fully meet PLOS ONE’s publication criteria as it currently stands. Therefore, we invite you to submit a revised version of the manuscript that addresses the points raised during the review process.

We look forward to receiving your revised manuscript.

Kind regards,

Itay Hen

Academic Editor

PLOS ONE

   "This work was supported by the Council for Science, Technology and Innovation (CSTI), Cross398 ministerial Strategic Innovation Promotion Program (SIP), Materials Integration for Revolutionary Design System of Structural Materials” (Funding agency: JST) and “Photonics and Quantum Technology for Society 5.0” (Funding agency: QST), and JST COI-NEXT Grant Number JPMJPF2221.

   This work was supported by JSPS KAKENHI (Grant Numbers JP21K03391, JP23H05447) and JST Grant Number JPMJPF2221. The Human Biology-Microbiome-Quantum Research Center (Bio2Q) is supported by the World Premier International Research Center Initiative (WPI), MEXT, Japan. 

    This work was supported by JST FOREST Program (Grant Number JPMJFR212K, Japan)."

    "This work was supported by the Council for Science, Technology and Innovation (CSTI), Cross-ministerial Strategic Innovation Promotion Program (SIP), Materials Integration for Revolutionary Design System of Structural Materials” (Funding agency: JST) and “Photonics and Quantum Technology for Society 5.0” (Funding agency: QST), and JST COI-NEXT Grant Number JPMJPF2221.

This work was supported by JSPS KAKENHI (Grant Numbers JP21K03391, JP23H05447) and JST Grant Number JPMJPF2221. The Human Biology-Microbiome-Quantum Research Center (Bio2Q) is supported by the World Premier International Research Center Initiative (WPI), MEXT, Japan.

This work was supported by JST FOREST Program (Grant Number JPMJFR212K, Japan)."

   "This work was supported by the Council for Science, Technology and Innovation (CSTI), Cross-ministerial Strategic Innovation Promotion Program (SIP), Materials Integration for Revolutionary Design System of Structural Materials” (Funding agency: JST) and “Photonics and Quantum Technology for Society 5.0” (Funding agency: QST), and JST COI-NEXT Grant Number JPMJPF2221.

This work was supported by JSPS KAKENHI (Grant Numbers JP21K03391, JP23H05447) and JST Grant Number JPMJPF2221. The Human Biology-Microbiome-Quantum Research Center (Bio2Q) is supported by the World Premier International Research Center Initiative (WPI), MEXT, Japan.

This work was supported by JST FOREST Program (Grant Number JPMJFR212K, Japan)."

6. Please amend either the title on the online submission form (via Edit Submission) or the title in the manuscript so that they are identical.

Reviewers' comments:

Reviewer's Responses to Questions

**Comments to the Author**

1. Is the manuscript technically sound, and do the data support the conclusions?

Reviewer #1: Yes

Reviewer #2: Partly

2. Has the statistical analysis been performed appropriately and rigorously? 

Reviewer #1: Yes

Reviewer #2: No

3. Have the authors made all data underlying the findings in their manuscript fully available?

Reviewer #1: Yes

Reviewer #2: No

4. Is the manuscript presented in an intelligible fashion and written in standard English?

Reviewer #1: Yes

Reviewer #2: No

5. Review Comments to the Author

Reviewer #1: Changes in representation for search problems has a history going back to the 1980s. The 'bumps' in the binary representation referred to by the authors, for instance, are known in this literature as 'Hamming cliffs'. The authors make no mention of this prior work.

A few key results in this literature are:

1. The 'no free lunch theorem' [Wolpert and Macready, IEEE Trans. Evol. Comp., 1(1), 67–82, 1997] and special-case corollaries [Whitley, Proc. 1st Ann. Conf. Gen. and Evo. Comp., 726–733, 1999].

2. A link between the advantage of the incremental shifts in a Gray or real representation and local correlations in the objective function ('schema similarity') [Caruana and Schaffer, Mach. Learn. Proc., 153–161, 1988].

3. Identification that the behaviour of a function is often highly scale dependent [Goldberg, Complex Systems, 5, 139–167, 1991], with search (often) rapidly approaching a regime in which Gray-type encoding is profitable, and the associated use of hybrid schemes involving dynamic representations, multi-step evolution, and redundant bits.

The representations proposed by the authors, including the randomised encodings, are similar to those considered in [Cheong et al., IEEE Symp. Found. Comp. Int., 251–258, 2007].

Now, with the above caveats, those aspects of the representation problem specific to the QUBO restriction and so to quantum annealing do remain, to my knowledge, largely unexplored. For instance, do Gray codes necessarily introduce higher order non-linearities that are disallowed in this setting? There appears to be room for discussion here, and I believe the results of the authors could serve to open this discussion, but some acknowledgement of prior, related work is necessary.

Otherwise, I believe the analysis in the paper is understandable and credible. The authors might consider slight revision to clarify their error analysis and to introduce some estimates of sample deviation in their plots.

Reviewer #2: Overall, this manuscript has many significant flaws, both in the presentation of the information, and the actual experimental details. This manuscript requires extremely extensive revision and justification of the proposed methods to be suitable for publication, and without significant improvement I would recommend rejection of the paper.

# General Notes

1. Figures 2, 3, 4, 5 do not have axis labels! They absolutely need axis labels for the figures to be understandable.

2. The primary focus of the paper is around what the authors refer to as "representation of real numbers". I find this particular phrase extremely confusing. The real numbers that are of interest here are the cost function values for discrete combinatorial optimization problems. First, they do not need to be high precision real numbers - they can be integers. But more importantly, I do not think that the proposed methods are representing real numbers in a different way, I think they are representing cost functions in a different way. Which is fine, assuming the cost functions are correct, but this term is very confusing.

3. Following from the previous point, lines 38-41 state that "we noticed that when real numbers are represented in binary number, there are numbers that can only be reached by inverting several bits simultaneously under the restriction of not increasing a given Hamiltonian, which makes the optimization very difficult.". I do not understand what is being said here -- what does it mean that real numbers are represented in "binary number". For optimization, there is a cost function that defines a mapping from a set of variables to some cost output, which is a real number. But, that does not mean that the real numbers are represented in binary, rather the binary variables have some cost defined by them depending on what cost function you care about. Also, why is the restriction of fewer bitflips relevant? Especially for quantum annealing, early on the anneal when the transverse field is dominating the states of the qubits are flipping very often. Additionally, bit assignments to get to a lower energy state do not necesarely need to be inverted all at the "same time".

4. The authors state that experiments were performed on D-Wave quantum annealers, and "Fixstars Amplify Annealing Engine". First, the exact names (e.g. chip ids in the case of the D-Wave quantum annealers) needs to be specified. Second, the technology of the devices needs to be described. For example, were any hybrid quantum annealing solvers used? What what is the technology behind "Fixstars Amplify Annealing Engine"? Much of the paper descrbed quantum annealing technology, but is "Fixstars Amplify Annealing Engine" a digital annealer, simulated annealing, some other type of classical optimization technique (and who is the manufacturer)? Third, the solver parameters used need to be very explicitely stated for all machines - for example, for the D-Wave quantum annealers was minor-embedding used, what annealing time was used, if minor embeddings were used what was the structure and number of variables, and what chain break resolution algorithm was used?

5. Line 151-152 states that "Since the longer the computation time, the better the expected solution, the timeout time". Well, this is not strictly true for all problem types in the case of quantum annealing.

6. Something to note is that D-Wave quantum annealers have limited precision for programming the coefficient weights on the hardware, so the uniform random coefficient choices will mean that the quantum annealers are unlikely to find the global ground state, especially if minor embedding was used (I assume it was).

7. Is equation 5 applied iteratively in order to compute better and better x solutions?

8. What is the matrix size that corresponds to the varying number of variables (line 237)? I assume it is the square root of N

9. Is q the decision variable in equations 3 and 4?

10. Line 147 and equation 5; I would like to see a more explicit example of one of these QUBOs -- I think that almost always QUBOs constructed for matrix factorization will yield higher order terms.

11. What are the "real variables x" in line 144? Are these the decision variables? or is x the coefficient of the decision variables?

12. i is not defined in equation 5

13. Line 139; what is {F_i, r} ? Are these two distinct real numbers being multiplied?

14. Critically, the central procedure that this paper describes is completely not clear. Equation 4 appears to be a polynomial containing only linear terms, with decision variables q. It is not clear how the four different "real number representations" are actually applied to the problem QUBO. To me, Equations 7, 8, 9, and 10 appear to just be four different ways of generating coefficients for the linear terms in Equation 4 -- it is completely unclear whether the generated QUBOs (all of which are, it seems, general integer matrix factorization QUBOs) using these four "real number representations" are actually equivalent (e.g. do the QUBOs/ Ising models have the same eigenspectrum, or at least the same ground states). This comparison only works (the central argument of the paper) if the four "real number representation" methods are actually producing equivalent QUBOs.

15. Line 166; If I understand correctly, the selected matrix coefficients are positive random, real numbers (up to some precision). If that is correct, then I am very curious what the exact QUBO formulation is (e.g. item 10), because these generated QUBOs would, I think, be very complex in terms of range of coefficients and total number of variables. And, it seems like some matrix entries are also chosen which are negative - which makes me even more curious what the exact QUBO formulation is, since I am not sure that matrix factorization with negative coefficients can be formulated as easily as is shown in eq. 5.

16. I suppose I would consider this a small comment, but it would be of interest to sample the problem QUBOs using classical heuristics as well, such as simulated annealing, steepest gradient descent, etc.

# Grammar / Spelling / clarification Notes

1. There are numerous grammatical errors throughout the maniscript that would need to be corrected before the paper is suitable for publication. Here are a few examples (this list is not comprehensive, and there are many more errors that need to be corrected):

1. Line 32. A potential fix would be to pluralize the phrase "quantum annealing machine"

2. Line 39. A potential fix would be "represented in their binary form"

3. Lines 56, 57, 58 do not make any sense - what does it mean to replace the objective function with the energy of the system? In quantum annealing specifically, the energy is the expectation value of the objective function, encoded using analog computation.

4. Lines 60, 61, 62 state that: "The principle of operation is to find a solution to the variables by adjusting quantum fluctuations so that the objective function eventually becomes the global minimum". This is not accurate, and also does not make sense - what does it mean for the objective function to become the global minimum? The objective function just tells us what the cost value, or energy, or expectation value is of the objective function for a specific set of spin (or variable) states.

5. Line 66 states that "Many differential equations are derived from the energy functional based on the principle of least action" - what does differential equations implicately say about why quantum annealing makes sense as an analog computation for physical systems? This reasoning about why quantum annealing may be useful as an analog computation is not a correct summarization of how quantum annealing works.

6. Lines 62 and 63: "Since the Ising model is used as the basis, binary variables are used" - this is incorrect, an ising model corresponds to spin variable states not binary variables.

7. Line 151: "the minimum Hamiltonian is explored using an Ising machine" I think the authors meant to say that the minimum variable assignment is computed using an Ising machine.

# Literature references

There are several papers relevant to the topic of solving (or obtaining good solutions) types of linear algebra problems using quantum algorithms, including quantum annealing, that the authors do not cite or discuss.

1. https://arxiv.org/abs/2206.10576

2. https://arxiv.org/abs/1704.01605

3. https://arxiv.org/abs/2007.05565

4. https://arxiv.org/abs/2205.00645

Note that the generalization of matrix factorization is tensor factorization, the boolean version of which has also been investigated in the context of quantum annealing:

1. https://arxiv.org/abs/2107.13659

2. https://arxiv.org/abs/2103.07399

3. https://ieeexplore.ieee.org/document/9325388

The authors make some numerous observations about bit-flipping of some variable states on the problem Hamiltonians, including simultaneous bit flipping. This paper which investigates this topic in quantum annealing may be of interest: https://arxiv.org/abs/2210.16513

Lastly, it seems to me that this manuscript falls somewhat into the category of problem Hamiltonian encoding, and there are a couple of papers on ideas for other types of encodings:

1. https://arxiv.org/abs/1903.05068

2. https://ieeexplore.ieee.org/abstract/document/9951263/

3. https://arxiv.org/abs/2108.12004

4. https://arxiv.org/abs/2102.12224

6. PLOS authors have the option to publish the peer review history of their article (what does this mean?). If published, this will include your full peer review and any attached files.

Reviewer #1: No

Reviewer #2: **Yes: **Elijah Pelofske

---

## [Author Response · Author response to Decision Letter 0]

17 Jan 2024

Response to the Reviewers’ Comments

Ref. #. PONE-D-23-25182

Title : Novel real number representations in Ising machines and performance evaluation: Combinatorial random number sum and constant division

Author : Katsuhiro ENDO, Yoshiki MATSUDA, Shu TANAKA and Mayu MURAMATSU

Responding to the review results, we have significantly revised the manuscript. We highly appreciate the reviewers’ valuable comments and suggestions for corrections, which make the quality of the paper improved.

In the response letter, we provide our responses to the reviewers’ comments. Since two reviewers gave their own comments separately, we have prepared separate answers. For each comment, we first cited reviewers’ remarks for reference, and then described how we have responded to those. The responses were reflected in the revised manuscript, in which the corrected or newly added contexts were written in red.

We would be very grateful if our paper could be published in “PLOS ONE”

Best regards,

Katsuhiro ENDO, Yoshiki MATSUDA, Shu TANAKA and Mayu MURAMATSU

---

## [Decision Letter · Decision Letter 1]

4 Mar 2024

PONE-D-23-25182R1Novel real number representations in Ising machines and performance evaluation: Combinatorial random number sum and constant divisionPLOS ONE

Dear Dr. Muramatsu,

Thank you for submitting your manuscript to PLOS ONE. After careful consideration, we feel that it has merit but does not fully meet PLOS ONE’s publication criteria as it currently stands. Therefore, we invite you to submit a revised version of the manuscript that addresses the points raised during the review process.

 One of the reviewers has indicated that while addressing most of their comments further revision is required. 

We look forward to receiving your revised manuscript.

Kind regards,

Itay Hen

Academic Editor

PLOS ONE

Reviewers' comments:

Reviewer's Responses to Questions

**Comments to the Author**

1. If the authors have adequately addressed your comments raised in a previous round of review and you feel that this manuscript is now acceptable for publication, you may indicate that here to bypass the “Comments to the Author” section, enter your conflict of interest statement in the “Confidential to Editor” section, and submit your "Accept" recommendation.

Reviewer #2: (No Response)

2. Is the manuscript technically sound, and do the data support the conclusions?

Reviewer #2: Partly

3. Has the statistical analysis been performed appropriately and rigorously? 

Reviewer #2: Yes

4. Have the authors made all data underlying the findings in their manuscript fully available?

Reviewer #2: Yes

5. Is the manuscript presented in an intelligible fashion and written in standard English?

Reviewer #2: No

6. Review Comments to the Author

Reviewer #2: # Follow up from my previous comments

I believe that the authors have, with great detail, addressed the majority of the questions and comments -- and I think that the current manuscript is significantly clearer. The clarifications on the real number representations make much more sense now, and the descriptions of quantum annealing theory and implementation are much clearer.

- For the literature references I suggested, to be clear the reason I mentioned matrix factorization papers is because they are solving types of linear equations, granted typically the underlying decision variables are spins or boolean variables.

- I do not think the authors need to perform such simulations, but I mentioned steepest gradient descent as a possible additional classical heuristic - and it is true that gradient descent does not actually apply for these problems, but there does exist discrete steepest descent solvers: https://docs.ocean.dwavesys.com/projects/greedy/en/latest/

- Additionally, now that the authors have clarified that the GPU based simulator was in fact running simulated annealing, I am satisfied with this comparison.

# Comments on the revised manuscript

1. An overall comment on the motivation and experimental description of the manuscript is that simulated annealing, and in general classical heuristics, are not constrained to perform only Hamming distance 1 (e.g. single bit-flip) local neighborhood searches. In particular, statements such as Line 351 where a single bit flip example is used, or Line 420: "bit-flip-based search, which mimics the process of simulated annealing" are not true that this type of search would mimic simulated annealing.

2. Line 84: "For example, consider the case where a single real number is optimized in a classical simulator": This example is unclear and should to be clarified -- what is being optimized? Is the real number the cost function?

3. Line 637: "The error bars denote the standard error of the mean errors." I do not understand this measure -- do the authors intend to say that the errors bars denote the standard deviation of the distribution of the error measure for the repeated trials?

4. Line 357: "Then, we search for the state that is closest to the target value, limiting the path to only repeated transitions to states with decreasing distances that can be reached by the flipping of bits up to the": What is the distance being used here? Is it hamming distance?

5. Line 395: "Fig 5 shows the results of the task, where each subplot in the figure corresponds to a different condition"; is this referring to each subplot showing results for a different condition number? Or, different parameter combinations? From looking at Figure 5, it does not appear that the condition number is mentioned in the plots.

6. There are still a few grammatical errors throughout the text that should be fixed.

7. PLOS authors have the option to publish the peer review history of their article (what does this mean?). If published, this will include your full peer review and any attached files.

Reviewer #2: No

---

## [Author Response · Author response to Decision Letter 1]

31 Mar 2024

Response to the Reviewers’ Comments

Ref. #. PONE-D-23-25182R1

Title : Novel real number representations in Ising machines and performance evaluation: Combinatorial random number sum and constant division

Author : Katsuhiro ENDO, Yoshiki MATSUDA, Shu TANAKA and Mayu MURAMATSU

Responding to the review results, we have significantly revised the manuscript. We highly appreciate the reviewers’ valuable comments and suggestions for corrections, which make the quality of the paper improved.

Below, we provide our responses to the reviewers’ comments. Since one reviewer gave his/her own comments separately, we have prepared separate answers. For each comment, we first cited reviewers’ remarks for reference, and then described how we have responded to those. The responses were reflected in the revised manuscript, in which the corrected or newly added contexts were written in red.

We would be very grateful if our paper could be published in “PLOS ONE”

Best regards,

Katsuhiro ENDO, Yoshiki MATSUDA, Shu TANAKA and Mayu MURAMATSU

[For Reviewer 2]

[Comment 1]

# Follow up from my previous comments

I believe that the authors have, with great detail, addressed the majority of the questions and comments -- and I think that the current manuscript is significantly clearer. The clarifications on the real number representations make much more sense now, and the descriptions of quantum annealing theory and implementation are much clearer.

- For the literature references I suggested, to be clear the reason I mentioned matrix factorization papers is because they are solving types of linear equations, granted typically the underlying decision variables are spins or boolean variables.

- I do not think the authors need to perform such simulations, but I mentioned steepest gradient descent as a possible additional classical heuristic - and it is true that gradient descent does not actually apply for these problems, but there does exist discrete steepest descent solvers: https://docs.ocean.dwavesys.com/projects/greedy/en/latest/

- Additionally, now that the authors have clarified that the GPU based simulator was in fact running simulated annealing, I am satisfied with this comparison.

[Response 1]

We thank the reviewer for taking the time to further review our manuscript and providing a critical assessment of our work. We also thank the reviewer for acknowledging most of our revisions to the previous peer review. Your comments, including those mentioned above, have deepened our peripheral knowledge of our research results. 

Regarding the papers on matrix factorization that the reviewer mentioned in the previous comment, they certainly solve linear equations, although their decision variables are binary variables. In future research, we would like to investigate the effect of real number representations in such matrix decomposition methods when the variables are real numbers represented by binary variables. According to your comment, we modified the sentence “They are often applied to combinational optimization problems [21] involving the objective functions of the target such as community detection [22], the traveling-salesman problem [23], and so on [24,25,26,27,28,29,30,31,32,33].” to “They are often applied to combinational optimization problems [21] involving the objective functions of the target such as community detection [22], the traveling-salesman problem [23], matrix factorization [24,25,26,27,28] and so on [29,30,31,32,33].” on page 3, line 54.

Regarding the steepest descent solvers, as the reviewer stated, it is sure that a discrete analogue of the steepest descent algorithm such as dwave-greedy can be applied to our binary optimization problems. When the naive steepest descent solver is applied to the linear equation problems, the computational accuracy achieved by each real number representation will likely be similar to our results. That is, the CONSTANT representation will reach the best accuracy that the representation can reach, while the BINARY representation will get stuck in a local solution. This is because the steepest descent solver is a method that reaches a local optimal solution, and the binary optimization problems that require flipping multiple bits simultaneously to reduce the value of the Hamiltonian are likely to be very complex nonconvex functions. This explanation has been added shortly to the text as follows on page 5, line 95: “A similar property can be inferred for the greedy steepest descent algorithm [47], which is another classical method, because of its repeated local optimization.”

The distance here is the Euclidean distance between the real value which is expressed in the binary variables using the real number representation and the target real value.

We have responded to all comments we have received this time, and we hope that you will confirm the corrections.

[Comment 2]

# Comments on the revised manuscript

1. An overall comment on the motivation and experimental description of the manuscript is that simulated annealing, and in general classical heuristics, are not constrained to perform only Hamming distance 1 (e.g. single bit-flip) local neighborhood searches. In particular, statements such as Line 351 where a single bit flip example is used, or Line 420: "bit-flip-based search, which mimics the process of simulated annealing" are not true that this type of search would mimic simulated annealing.

[Response 2]

Thank you for your comments regarding the simulated annealing method. As you pointed out, local neighborhood search with Hamming distance of 1 is not a strictly necessary constraint in simulated annealing. However, for example, in the case of the simulated annealing method that utilizes the Metropolis-Hastings algorithm (which is commonly used as an implementation of simulated annealing), if the state is changed drastically at a single step, the energy difference between the states before and after the change becomes too large, resulting in a very low probability of movement. For this reason, it is certain that local neighborhood search, in which only one part of the state is changed, is often employed for efficiency reasons.

In accordance with your comment, we have revised the manuscript. We added the following sentences on page 16, line 355:

In Tasks 1 and 2 we are considering the repeated process of moving to a such neighborhood. The local neighborhood search with Hamming distance of 1 is not a strictly necessary constraint in simulated annealing. However, for example, in the case of the simulated annealing method that utilizes the Metropolis-Hastings algorithm (which is commonly used as an implementation of simulated annealing), if the state is changed drastically at a single step, the energy difference between the states before and after the change becomes too large, resulting in a very low probability of movement. For this reason, it is certain that local neighborhood search, in which only one part of the state is changed, is often employed for efficiency reasons.

We also modified the sentences “In Task 1 and 2, we examined the efficiency of bit-flip-based search, which mimics the process of simulated annealing when using different real number representations.” to “In Tasks 1 and 2, we examined the efficiency of bit-flip-based search, which mimics the process of simulated annealing with local neighborhood searches when using different real number representations.” on page 20, line 434.

[Comment 3]

2. Line 84: "For example, consider the case where a single real number is optimized in a classical simulator": This example is unclear and should to be clarified -- what is being optimized? Is the real number the cost function?

[Response 3]

We apologize for the misleading wording which contained a grammatical error. The correct phrase is “Consider solving an optimization problem with only one real number as the decision variable”. Following your comment, we modified our manuscript. We modified the sentence “For example, consider the case where a single real number is optimized in a classical simulator of annealing machine.” to “For example, consider solving an optimization problem with only one real number as the decision variable in a classical simulator of annealing machine.” on page 4, line 84.

[Comment 4]

3. Line 637: "The error bars denote the standard error of the mean errors." I do not understand this measure -- do the authors intend to say that the errors bars denote the standard deviation of the distribution of the error measure for the repeated trials?

[Response 4]

Thank you for pointing out the part where the wording is not appropriate. This error bar shows the standard error of the estimated mean value of absolute error E (Eq. 11 in our manuscript). In this experiment, we have performed multiple trials with the same conditions for statistical analysis, and each trial returns a single error E. Based on the error E which varies stochastically in each trial, the mean value of error is estimated. The error bar shows the standard error, i.e., error of estimated value. The error bars in Fig. 2 are sufficiently small that the mean of the errors can be considered sufficiently reliable.

For better clarity, we have revised the text. We modified the sentence “The error bars denote the standard error of the mean errors.” to “The error bars denote the standard error of the estimated mean errors.” on page 31, line 653.

[Comment 5]

4. Line 357: "Then, we search for the state that is closest to the target value, limiting the path to only repeated transitions to states with decreasing distances that can be reached by the flipping of bits up to the": What is the distance being used here? Is it hamming distance?

[Response 5]

I apologize for not being clear about the distance in the part you mentioned. The distance here is the Euclidean distance between a real value, which is expressed in the binary variables using a real number representation, and a target value, which is also a real number. In other words, in Task 1 on line 357, one real value expressed in the binary variables using any of the four real number representations is allowed to move only in the direction closer to the target value. This explanation has been added to the text as follows on page 17, line 370:

The distance here is the Euclidean distance between the real value which is expressed in the binary variables using the real number representation and the target real value.

[Comment 6]

5. Line 395: "Fig 5 shows the results of the task, where each subplot in the figure corresponds to a different condition"; is this referring to each subplot showing results for a different condition number? Or, different parameter combinations? From looking at Figure 5, it does not appear that the condition number is mentioned in the plots.

[Response 6]

Thank you for pointing out the unclear statement in the relevant section. Fig. 5 shows the results of the 10000 trials each with different values of K (the number of steps) and R (the number of bits). Each subplot in Fig. 5 (except for the statistical summary on the left) corresponds to the results of specific combinations of K and R. In other words, the "different condition" in L. 395 refers to a variation in both K and R. The values of K and R for each subplot are listed in the subtitle at the top of each subplot.

Note that in Task 2, which includes L. 395, there is no condition number because we are considering a simplified problem with only one real number as the decision variable. In response to your comment, we have revised the text as follows to clarify exactly what a "different condition" is. We added the following sentence on page 17, line 376: The condition of each subplot (the values of K and R) is listed in the subtitle at the top of each subplot.

[Comment 7]

6. There are still a few grammatical errors throughout the text that should be fixed.

[Response 7]

We apologize for the remaining grammatical errors. We have sent the manuscripts to English proofreading services, and the revised manuscript reflects the corrections.

As a final note, we would like to express our appreciation for your careful recommendations and thorough review of our paper. If you have any additional concerns that we can address to improve our paper, please do not hesitate to let us know

---

## [Editor Report · Decision Letter 2]

15 May 2024

Novel real number representations in Ising machines and performance evaluation: Combinatorial random number sum and constant division

PONE-D-23-25182R2

Dear Dr. Muramatsu,

We’re pleased to inform you that your manuscript has been judged scientifically suitable for publication and will be formally accepted for publication once it meets all outstanding technical requirements.

Kind regards,

Itay Hen

Academic Editor

PLOS ONE

---

## [Editor Report · Acceptance letter]

4 Jun 2024

PONE-D-23-25182R2 

PLOS ONE

Dear Dr. Muramatsu, 

I'm pleased to inform you that your manuscript has been deemed suitable for publication in PLOS ONE. Congratulations! Your manuscript is now being handed over to our production team.

Kind regards, 

on behalf of

Dr. Itay Hen 

Academic Editor

PLOS ONE